# Contrasted Epidemiological Patterns of West Nile Virus Lineages 1 and 2 Infections in France from 2015 to 2019

**DOI:** 10.3390/pathogens9110908

**Published:** 2020-10-30

**Authors:** Cécile Beck, Isabelle Leparc Goffart, Florian Franke, Gaelle Gonzalez, Marine Dumarest, Steeve Lowenski, Yannick Blanchard, Pierrick Lucas, Xavier de Lamballerie, Gilda Grard, Guillaume André Durand, Stéphan Zientara, Jackie Tapprest, Grégory L’Ambert, Benoit Durand, Stéphanie Desvaux, Sylvie Lecollinet

**Affiliations:** 1UMR 1161 Virology, ANSES, INRAE, ENVA, ANSES Animal Health Laboratory, EURL for Equine Diseases, 94704 Maisons-Alfort, France; gaelle.gonzalez@anses.fr (G.G.); marine.dumarest@anses.fr (M.D.); steeve.lowenski@anses.fr (S.L.); stephan.zientara@anses.fr (S.Z.); sylvie.lecollinet@anses.fr (S.L.); 2National Reference Laboratory for Arboviruses, Institut de Recherche Biomédicale des Armées, 13010 Marseille, France; isabelle.leparc-goffart@inserm.fr (I.L.G.); Gilda.Grard@inserm.fr (G.G.); guillaume.durand@inserm.fr (G.A.D.); 3Unité des Virus Emergents (UVE: Aix Marseille Univ, IRD 190, INSERM 1207, IHU Méditerranée Infection), 13385 Marseille, France; xavier.de-lamballerie@univ-amu.fr; 4Regional Office Paca-Corse, Santé Publique France, 13331 Marseille, France; Florian.franke@santepubliquefrance.fr; 5ANSES Ploufragan-Plouzane-Niort, Viral Genetics and Biosecurity Unit, 22440 Ploufragan, France; Yannick.blanchard@anses.fr (Y.B.); pierrick.lucas@anses.fr (P.L.); 6ANSES Animal Health Laboratory, PhEED Unit, Dozulé site, F-14430 Goustranville, France; Jackie.tapprest@anses.fr; 7EID Méditerranée, 34184 Montpellier, France; glambert@eid-med.org; 8Epidemiology Unit, Paris-Est University, ANSES Animal Health Laboratory, 94704 Maisons-Alfort, France; benoit.durand@anses.fr; 9Office Français de la Biodiversité, Unité Sanitaire de la Faune, 01330 Birieux, France; stephanie.desvaux@ofb.gouv.fr

**Keywords:** arbovirus, emerging infectious diseases, zoonotic, West Nile, lineages 1 and 2, France

## Abstract

Since 2015, annual West Nile virus (WNV) outbreaks of varying intensities have been reported in France. Recent intensification of enzootic WNV circulation was observed in the South of France with most horse cases detected in 2015 (*n* = 49), 2018 (*n* = 13), and 2019 (*n* = 13). A WNV lineage 1 strain was isolated from a horse suffering from West Nile neuro-invasive disease (WNND) during the 2015 episode in the Camargue area. A breaking point in WNV epidemiology was achieved in 2018, when WNV lineage 2 emerged in Southeastern areas. This virus most probably originated from WNV spread from Northern Italy and caused WNND in humans and the death of diurnal raptors. WNV lineage 2 emergence was associated with the most important human WNV epidemics identified so far in France (n = 26, including seven WNND cases and two infections in blood and organ donors). Two other major findings were the detection of WNV in areas with no or limited history of WNV circulation (Alpes-Maritimes in 2018, Corsica in 2018–2019, and Var in 2019) and distinct spatial distribution of human and horse WNV cases. These new data reinforce the necessity to enhance French WNV surveillance to better anticipate future WNV epidemics and epizootics and to improve the safety of blood and organ donations.

## 1. Introduction

West Nile virus (WNV) is an arthropod-borne flavivirus transmitted by the bites of infected mosquitoes mostly belonging to the *Culex* genus [1]. WNV is maintained in an enzootic bird-mosquito cycle but can be transmitted through mosquito bites to dead-end hosts, such as humans or equids and occasionally cause neuro-invasive disease that can be lethal in these hosts [2]. In Europe, WNV outbreaks occur during the summer and fall seasons (July–October) when *Culex* mosquitoes are abundant.

According to phylogenetic analysis, eight different WNV lineages have been described [3]. WNV lineages 1 and 2 are the most widespread and caused most of the major epidemics encountered so far [4]. WNV lineage 1 was first reported in Europe in the 1960s when seropositive animals (horses and cattle) or viral isolates (mosquitoes and humans) were identified in France, Portugal, and Cyprus [5,6]. After more than 30 years without experiencing WNV outbreaks, North African, Western, and Eastern European countries reported again the emergence of WNV lineage 1 strains belonging to the Western-Mediterranean clade (in North Africa and Western Europe such as in Morocco in 1996, Italy in 1998, and in France in 2000) [7,8,9] and the Eastern-European clade (Eastern Europe during the 1996 Romanian outbreak and Russia in 1999) [10] affecting mainly equids and or/humans respectively. In France, after the 2000 WNV outbreaks in the Camargue area, sporadic cases of West Nile fever (WNF) occurred in departments bordering the Mediterranean coast in the 2000s (Var in 2003, Bouches-Du-Rhône, Gard and Hérault in the Camargue area in 2004, and Eastern Pyrenees in 2006) [11,12]. WNF has been mainly reported in France in horses, while human cases have been less frequently observed (seven cases in Var in 2003) [13].

Most European WNV outbreaks before 2010 had been caused by WNV lineage 1 strains. Unexpectedly, recent increase in WNV transmission and outbreaks, noticeable in Europe since 2010, has been associated with the introduction and spread of WNV lineage 2 strains [14,15,16]. A first WNV lineage 2 strain was initially detected in Hungary in 2004 [17] and subsequently spread to the eastern part of Austria in 2008 [15,18], to the Balkan peninsula, including Greece in 2010 [19], Serbia, Croatia, and Bulgaria in 2012 [20,21], further East to Italy in 2011 [22], and more recently it reached Spain in 2017 [23] and Germany in 2018 [24,25]. Another WNV lineage 2 strain, first detected in 2004 in Rostov Oblast in Southern Russia [26], has also been occasionally reported in Europe, in Romania [27] in 2010, in Italy in 2014 [28], and in Greece in 2018 [29].

After WNV reemergence in the Camargue area in 2000, a multidisciplinary WNV monitoring system has been implemented in France since 2001 including clinical surveillance in wild birds, horses, and humans, and records of mosquito abundance and diversity during the transmission season. Such clinical surveillance is implemented in departments from the Mediterranean area during the WNV at-risk period from 1 June until the end of November [11].

Here we report recent French WNV outbreaks (2015–2019) in humans, equids, and the wild avifauna in the Mediterranean area and describe the emergence of WNV lineage 2 in France in 2018 and the changing patterns of infection in humans and horses following this emergence.

## 2. Results

### 2.1. Comparison of WNV Seasonal Patterns in France between 2015 and 2019

#### 2.1.1. WNV Outbreaks in France in 2015–2019

In France during the last five years, the most important equine WNV outbreaks were reported in 2015. Of the 49 cases reported in horses in 2015, 41 presented neuroinvasive forms, three febrile forms, and five asymptomatics were identified thanks to serosurveys implemented in identified WNV transmission foci [11]. There was no equine case noticed in 2016, while one asymptomatic horse was detected in 2017 in the vicinity of WNV human cases. A total of 13 and nine neuroinvasive cases were reported in 2018 and 2019 respectively, while four additional febrile forms were reported in 2019 (Figure 1). Natural death or euthanasia occurred in 14.6% (6/41) and 15.4% (2/13) of the horses with West Nile neuroinvasive disease (WNND) in 2015 and 2018 respectively whereas no equine death was reported in 2019. These percentages are lower than usually described in the literature [30].

For the same period, for humans, the highest number of autochthonous cases was reported in 2018 with 26 laboratory-diagnosed human cases, including seven WNND, 18 febrile and one asymptomatic form. Interestingly, one blood donor, symptomatic a few days after the donation and one organ asymptomatic donor were tested positive for WNV the same year [31]. The total number of cases in 2018 represented a 27-fold increase compared with the 2015–2017 transmission seasons during which one, none, and two febrile cases were reported in 2015, 2016, and 2017 respectively. In 2019, one febrile and one WNND were reported (Figure 1). WNND have generally occurred more frequently in horses than in humans in France so far.

For the first time in France since the implementation of an integrated WNV surveillance system, WNV infections were also reported through WNV surveillance in the avifauna in 2018. In total, four raptors (two northern goshawks (*Accipiter gentilis*), one common buzzard (*Buteo buteo*), and one long-eared owl (*Asio otus*) were diagnosed WNV positive in September and October 2018 in Corsica (owl) and Alpes-Maritimes (diurnal raptors) by the SAGIR network (a French network dedicated to wildlife disease surveillance). All these wild birds were found alive and suffered from serious nervous disorders.

#### 2.1.2. Shifts in Temporal and Spatial Distribution of WNV Cases

In 2015, the onset of the first equine case was reported on week 33 (starting the 11 August) with a peak on week 38 (14 September to 20 September) and the last case was notified on week 44 (Figure 2a).

Each of the 49 equine cases were identified in three departments surrounding the Camargue area, a large region delineated by the Rhone delta and characterized by high biological and environmental diversity in South-Eastern France. A total of 33 confirmed cases corresponding to 26 distinct outbreaks were located in Bouches–du-Rhône department, 15 confirmed cases (12 outbreaks) in Gard department, and one in Hérault department. Only one WNF human case was confirmed later in the season (2 October 2015, onset of symptoms 27 September) in Gard department and one mosquito pool corresponding to *Culex pipiens* mosquitoes was found positive in the same area on 11 September 2015 (Figure 3c,d).

In 2017, two human cases were diagnosed on the 21 August and 4 September in Alpes-Maritimes department. It was the first time in France that WNV was reported in this area. Following these cases, a serosurvey was carried out on 151 equids from a horse center located in the vicinity of the second WNV human case. Only one asymptomatic horse (1/151; CI 0%−2.9%) was found WNV-IgM positive (Figure 3e,f).

The WNV transmission season started much earlier and finished later in 2018 compared to the 2015–2017 period. Even if horses are particularly sensitive to the infection and can be used as indicators of virus circulation [32], autochthonous human cases were diagnosed before the occurrence of horse cases from 19 July 2018, while symptoms onset dated back to early July (week 27, 2–8 July). The number of human cases peaked on week 33 and the last case was notified week 46 (12–18 November) (Figure 2b). Four French departments around the Mediterranean Sea reported WNV infections (i.e., Alpes-Maritimes, Bouches-du-Rhône, Vaucluse, and Eastern Pyrenees). Alpes-Maritimes department was the first area reporting WNV cases and a cluster of 21 cases were located exclusively in this area (Figure 3h). Three raptors were found WNV positive later on in the season in the same area (in Nice and Antibes, Figure 3g). For the first time in 2018, an outbreak occurred in the south of Corsica with the first human case diagnosed at week 32 (6–12 August) and a second one at week 39 (24–30 September). Concomitantly, the onset of WNV equine outbreaks was reported on week 35 (starting 27 August) in North Corsica (Bastia) and four horses were found positive in 2018 in Corsica (Figure 3g). Finally, one long-eared owl (*Asio Otus*) was also diagnosed positive in South Corsica. Interestingly, the other 2018 equine cases were located in two departments in the Camargue area, namely Bouches-du-Rhône and Hérault, already affected by WNV equine outbreaks in 2015 (Figure 3g). The notification of WNV cases in equids started later than in humans and at a comparable period than in 2015 (week 33, 11 August 2015 versus week 35, 29 August 2018).

WNV activity in 2019 was lower than in 2018. It was characterized by a circulation of the virus in the Camargue region with WNV infected horses (*n* = 11) and by the resurgence of WNV disease in Corsica (n = 2 horses). Moreover, one WNF and one WNND human cases were detected in Var, a department with sporadic WNV transmission to humans and horses identified since 2003 [33].

Recent changes in the temporal and spatial distribution of French WNV cases can be highlighted from 2015–2019 data analysis. Specifically, a multiplication of circulation foci have been reported during the last three years, with the emergence of WNV and recent description of clinical cases in the departments of Alpes-Maritimes, Var, and French Corsica island alongside the usual enzootic WNV circulation in the Camargue area during most of the period (2015, 2018–2019). WNV emergence in Alpes-Maritimes was associated with an increase of reported WNV cases in humans and birds but not in equids, while the distinct spatial distribution of human (mostly in Alpes-Maritimes) and horse (mostly in Camargue) WNV cases have been observed in Southern France these last years.

The intensity of WNV circulation and transmission is shaped in part by mosquito vector abundance and is influenced by biotic and abiotic factors favorable for mosquito proliferation [34,35]. Mosquito abundance was found to be significantly lower in Hérault than in the other two departments for the period 2015 to 2019 but the number of traps was six for this department compared to eight for the Bouches-du- Rhône and Gard (see Section 4.2 material and methods). The analysis of *Culex pipiens* mosquito abundance in the Camargue area (Bouches-Du-Rhône, Gard, and Hérault) during the vector season (June to October) indicated that the total number of trapped mosquitoes was significantly higher in 2018 than in 2016, 2017, and 2019 (*p* values ≤ 0.002). The difference observed between 2015 and 2018 was not found significant (Figure 4b and Table 1). Moreover, since the importance of WNV outbreaks in Europe was found to be strongly correlated with the length of the mosquito proliferation season, early abundance of mosquitoes in June was compared in 2015–2019. The vector season started earlier in 2018 than in other years, as the number of mosquitoes trapped in June was significantly higher in 2018 than in 2015, 2016, 2017, and 2019 (*p* values < 0.0007) (Table 1). The abundance ratio for June 2015 was 0.20, thus corresponding to an abundance five times higher in June 2018 than in June 2015, after controlling for the effect of the department (Figure 4a, Table 1).

### 2.2. First Description of WNV Lineage 2 Isolates in France in 2018

Phylogenetic analysis of the virus isolated from the brain of a WNV-infected horse in 2015 identified a lineage 1 strain belonging to the Western Mediterranean clade and genetically related to earlier French isolates collected in the Camargue area in 2000 and 2004 (Figure 5). It suggests an endemic circulation of the virus in the Camargue area, with WNV cycling in most years 2000–2014 between birds and *Culex* mosquitoes only and spilling over to horses and humans more regularly in 2015–2019.

Interestingly, WNV lineage 2 was recovered from raptor specimens found moribund in Alpes-Maritimes in 2018, demonstrating a recent emergence of WNV lineage 2 in Southeastern France. WNV strains showed the highest genetic homology with WNV strains reported recently in 2014 in Northern Italy (Veneto and Lumbardy) (Figure 5).

WNV strains detected in France, in one horse in 2015 (WNV-Akela/France/2015, indicated by a circle) and in wild birds in 2018 (WNV-6125/France/2018 and WNV-7025/France/2018, highlighted with triangles) belonging to different lineages, with a homology of 79.7–79.8% at the nucleotide level and 93.9–94.0%% (3223/3435) at the amino acid level (Appendix A). 9–14, and eight amino acid substitutions affecting different viral genes were observed between WNV-Akela/France/2015 and older French lineage 1 isolates and between lineage 2 WNV-7025/France/2018, and the closely genetically related WNV-Cremona4/Italy/2014 respectively (Appendix A). No amino acid substitutions correspond to established WNV molecular virulence determinants or to positively selected codons [37,38].

## 3. Discussion

In France, WNV caused outbreaks involving several human and horse cases in the beginning of the 1960s before it disappeared for 35 years [7]. During the 2000–2008 period, four episodes of WNV transmission were reported in France. Only WNV lineage 1 was reported in France during this period, with closely genetically related WNV isolates belonging to the Western Mediterranean clade identified in the Camargue area in 2000 (in horses) [39] and in 2004 (in birds) (Figure 5) [40]. It re-emerged following a cyclical and hardly predictable pattern and was mostly limited to Camargue, a high-risk area for WNV circulation due to high concentration of wetlands, mosquitoes, wild birds, and horses [41,42,43]. The same spatial Camargue location of equine outbreaks was pointed out in 2015, 2018, and 2019. The enhanced abundance of WNV competent mosquitoes in 2015 and 2018 and an earlier vector season in the Camargue area in 2018 have been identified as potential risk factors for higher WNV transmission. Moreover, the phylogenetic analysis of one WNV strain identified recently in Camargue supports WNV enzootic transmission in this region as it revealed that the sequence of WNV isolated from a confirmed equine case in 2015 is close to the lineage 1 strain that circulated in France in 2004 [44,45]. The percentage of nucleotide identity between French WNV lineage 1 isolates (>98.3%, Appendix A) is coherent with the mean evolutionary rate of the European WNV strains (3.7 × 10^−4^ substitutions/site/year) [46]. Most of the mutations distinguishing the viral isolates were synonymous and homogenously distributed along the viral genome, which suggests that the genetic evolution of French WNV strains arose through a strong and local diversifying selection.

In 2018, WNV lineage 2 belonging to the Central and Eastern European clade (CEC) as defined by Ziegler et al. [47] was isolated for the first time from wild raptors, which have been shown to be particularly susceptible to WNV neuro-invasive infections, in Alpes-Maritimes in France [18,48,49]. WNV lineage 2 emergence in France was associated with exceptional WNV activity and lineage 2 spread in Western and Northern-most territories (Germany) in Europe this same year [25]. During this year, WNV infections in Europe increased dramatically compared to previous transmission seasons. From June to November 2018, a large part of Europe faced a period of unusually hot weather that led to record-breaking temperatures [50]. Like in France, European WNV infections started earlier in 2018 than in previous years. Indeed, the first WNF cases were reported on 31 May (week 22) in Greece which is the earliest disease onset compared with previous years [51]. At the end of 2018, a total of 1503 human infected confirmed cases were reported in 11 countries of the European Union (with almost 92% of cases coming from Italy, Greece, Romania, and Hungary) [52]. This number exceeded the cumulative number of WNV reported infections of the seven previous years [53]. The highest increase compared to previous transmission season was observed in Bulgaria (15 fold) followed by France (13.5 fold), and Italy (10.9 fold) [53]. During the 2018 transmission season, reports from the ECDC also indicated a high transmission among horses with 285 outbreaks reported by European member states as follows: 149 in Italy, 91 in Hungary, 15 in Greece, and 13 in France representing an increase of 30% in comparison with the number of outbreaks in 2017 [53].

In particular, in 2018, there was a large WNV lineage 2 outbreak in Northern Italy, including the Piemonte regions. WNV lineage 2 circulation was first documented in Italy in 2011 and, since then, has settled in Northern Italy at least since 2013 [14,54]. Recent phylogenetic analysis [55] revealed that two Italian lineage 2 strains, namely clade A and clade B diverged between 2010 and 2012 from a central region of the Po Valley. Clade A spread towards Northeastern Italy and apparently became extinct in 2013–2014, whereas clade B spread north-west reaching the most Western regions of Italy. Such a WNV short distance introduction via infected birds coming from a neighboring country has been usually hypothesized [18,47]. According to the high percentage of nucleotide homology (99.76%) between the Italian 2014 clade B and French 2018 WNV strains (Appendix A), we hypothesize that WNV lineage 2 gradually spread from Northern Italy to South-Eastern France in 2018 or 2017, considering that WNV lineage 2 was isolated from wild birds in Alpes-Maritimes in 2018 and that human cases were already reported in the same area in 2017. Moreover, Histidine instead of Proline residues are found at position 249 in the helicase part of NS3 in recent Italian and French lineage 2 strains. The role of this genetic modification in the modulation of WNV pathogenicity in mammalian and bird hosts has been regularly debated [56,57]. While a specific non-synonymous substitution, Lys2114Arg, was identified in several WNV lineage 2 isolates obtained in Germany in 2018 and could be associated with an increased WNV fitness as such a mutation was not evidenced in the French 2018 lineage 2 strains.

In 2018, most human cases in mainland France were reported in areas with WNV-related bird mortality, which is consistent with the positive relationship between WND human cases and seroprevalence level in passerine birds, previously demonstrated at the European level [58]. The absence of horse cases before the onset of human cases could be explained by the very low density of horses in the Alpes-Maritimes area and by the location of 13 out of 26 human cases in the urban city of Nice [59]. The increase observed in 2018 in WNV human cases in France could result from differing virulence or transmission properties for humans and for horses of WNV lineages 1 and 2 and from varying animal and human densities in areas reporting WNV infections in 2018 (Alpes-Maritimes with densely populated urbanized areas, while most horse cases were reported in a natural wetland, the Camargue area); the first hypotheses (with more transmission to and more cases in humans associated with WNV lineage 2 infections) would deserve more attention but are currently not supported by the literature.

We also document in 2018 the detection of WNV for the first time in French Corsica Island. This finding is not surprising as a serosurvey carried out in Corsica in 2014 highlighted that 9.4% of horses presented WNV antibodies. Among these positive horses, 66.6% were native from the island, indicative of WNV local circulation [60]. The identification of WNV clinical cases in humans, horses, and birds in Corsica further documents recent and active circulation of the virus. Nevertheless, the identification of the causative lineage could not be achieved as a low viral load was evidenced in clinical specimens collected on the island (one raptor, a long eared-owl) and as both lineages 1 and 2 were described recently in Sardinia and Italy [61].

Fewer outbreaks were reported in 2019 than in 2018 in France and Europe, but a changing epidemiological pattern of WNV circulation can be anticipated in France in the coming years. Indeed, the introduction of WNV lineage 2 in Hungary in 2004 was followed by strain adaptation and limited activity in 2005–2007 while extensive spread of the virus was reported from 2008 [18]. Moreover, a remarkable extension of the distribution area of WNV lineage 2 has been evidenced in 2018. Specifically equine cases and mortality on resident wild and captive birds were detected for the first time in Eastern and Southeastern Germany [25]. This introduction was followed in Germany by an increase of equine WNV outbreaks in 2019 and the reporting of the first five confirmed mosquito-borne autochthonous human cases [47]. Another important finding during the 2018 transmission season relates to WNV genome detection in one blood donor for the first time in France by the French blood establishment [31]. Interestingly this donor had spent time in Alpes-Maritimes before the occurrence of the first animal or human WNV cases. These new data on WNV spread in South France and on positive WNV screenings in blood products and organ transplants highlight the necessity to strengthen WNV integrated surveillance in France in order to primarily secure human blood, cells, or organ products. WNV surveillance has been mainly supported by clinical surveillance programs focusing on the analysis of moribund or dead birds and of horses and human patients with neuroinvasive signs, which may lack sensitivity and fail to detect low-level circulation. A combination of clinical event-based surveillance activities and active monitoring of WNV enzootic transmission through the regular monitoring of seroconversions in sentinel and/or resident birds and horses or through mosquito trapping and WNV screening would enhance the chance to early detect WNV transmission [62].

Finally, an enhanced transmission of WNV in Southeastern France in 2018 paralleled an unusually high number of outbreaks of another *Culex*-borne flavivirus, Usutu, and in most French metropolitan territories [63]. Such findings emphasize the need of unraveling the virological, ecological, and climatic factors responsible for *Culex*-borne flavivirus emergence in France and Europe [64,65].

## 4. Materials and Methods

### 4.1. Samples

Main organizational aspects of the French West Nile virus surveillance system in animals, humans, and vectors have been described previously in the article of Bahuon et al. [11]. Briefly, the surveillance is based on clinical case definition (human and equine) or criteria for dead or sick birds’ reports, collection, and testing. No routine indicator-based surveillance is implemented on the animal population. Diagnostic specimens are from 1/suspect human cases, as well as *Culex* mosquito populations sampled in affected areas once the viral circulation has been confirmed. They are analyzed by the National Reference center (NRC) for arboviruses (IRBA-Armed Forces Biomedical Research Institute) 2/each horse and avian suspect cases confirmed by the National Reference laboratory (NRL) on West Nile virus (Anses, Animal Health Laboratory, Maisons-Alfort) [12]. Suspect West Nile cases correspond to human patients over 15 years old and equids presenting with fever (≥38.5 °C) and symptoms of viral meningitis or encephalitis; wild or captive birds (raptors, corvids, and turdids more specifically) found dead, and individuals displaying neurological symptoms during the surveillance period (1 June to end of November) in the at-risk area (i.e., counties in the Mediterranean area). Moreover wild bird surveillance has been extended to departments considered, according to a statistical model, at an increased risk of WNV transmission, and located along the Mediterranean Sea and the Rhone River in South Eastern France, as well as in Bas-Rhin in North Eastern France since 2019 [58].

### 4.2. Mosquito Collection

Mosquitoes were collected weekly from mid-May to late October, corresponding to the mosquito season in the Rhône Delta, Camargue. CDC-like traps (John W. Hock Company, Gainesville, FL, USA) were used without light and were baited with carbon-dioxide dry ice (−80 °C). The trapping network was composed of 8 traps in the departments of Gard, 8 traps in Bouches-du-Rhône, and 6 traps in Hérault departments. Mosquitoes were stored in the fridge, killed, and identified with identification morphological keys.

### 4.3. Serology

Blood samples were collected in dry tubes, allowed to clot, and centrifuged at 1500 rpm for 10 min and stored at +4 °C during 1 month at most or at −20 °C for long term archiving.

For equine suspected cases reported to the French NRL, sera were first screened for anti-WNV antibodies by competition ELISA (ID Screen West Nile competition kit, IDVet Company, Montpellier, France) in local veterinary laboratories. Then IgG positive sera were further analyzed by M-antibody capture ELISA for IgM detection (ID screen West Nile IgM capture, IDVet company, Montpellier, France) in local veterinary laboratories and confirmed at the NRL. Analysis and interpretation of ELISAs were performed according to the manufacturer’s instructions. In the event of IgM positive screening, the first samples collected during WNV outbreaks were confirmed by microneutralization test (MNT) as described in Beck et al. [66]. A confirmed case was therefore defined as a clinical suspected horse with at least a positive IgM ELISA test.

For human WNV diagnosis, sera and cerebrospinal fluid (CSF) were tested by in-house ELISAs (indirect IgG and MAC-ELISAs) using precipitated and inactivated virus. A case of WNV infection is confirmed with the presence of IgM in CSF and/or IgM and IgG in sera and anti-WNV neutralizing antibodies [67].

### 4.4. Real-Time RT-PCR

Brain of horses and birds, EDTA blood, and CSF suspected to be infected with WNV were stored at −80 °C until analysis. Brains were grinded in Dulbecco modified Eagle’s minimal essential medium (DMEM) with ceramic beads (MP Biomedicals, Illkirch, France) and FastPrep ribolyzer in BSL3 facilities. A total of 560 µL of Lysis buffer from the QIAamp Viral RNA kit (Qiagen, Hilden, Germany) were added to 140 µL of grinded material before RNA extraction with the automate QIAcube. Human samples (EDTA blood and CSF) were processed the same way. Every RNA extracts were subjected to real time (rt) RT-PCR following the protocol described earlier by Linke et al. [68].

### 4.5. Virus Isolation

One milliliter of brain homogenates of WNV rtRT-PCR positive wild birds and horses was prepared in DMEM culture medium and inoculated on T25 flask that had been seeded with Vero NK cells (ATCC: CCL81™), 24 h earlier and washed with DMEM before inoculation. After 1h 30 of incubation at 37 °C with 5% CO_2_, cells were washed twice with phosphate buffered saline (PBS), and complete medium (DMEM+ 1% penicillin- streptomycin+ 1% sodium pyruvate + 5% fetal calf serum) was added. The cells were observed each day from 3 days to 7 days post infection (pi). As soon as cytopathic effects (CPE) were detected, the supernatant was collected, stored at −80 °C, and RNA extracts subjected to rtRT-PCR to confirm WNV detection. Primary isolation was followed by a passage on Aedes albopictus (C6/36) (ATCC^®^ CRL1660™) cell line. A total of 200µL−1 mL of Vero cell supernatants was added to T25 flask that had been seeded with C6/36 24 h earlier and washed with Leibowitz L15 medium before inoculation. After 1h 30 of incubation at 28 °C without CO_2_, 6 mL of Leibowitz L15 media + 1% penicillin- streptomycin+ 1% sodium pyruvate + 1% L Glutamin+ 10% fetal calf serum were added. CPEs were not systematically observed in C6/36 cells and supernatants were collected on day 7 post-infection at the latest and tested as described above. This protocol is adapted from the OIE Manual of Diagnostic Tests and vaccines for Terrestrial Animals [69].

### 4.6. Nucleotide Sequencing and Sequence Analysis

Sequencing libraries were prepared from genomic RNAs extracted from virus isolate (2015) or from organ homogenates (2018) and whole-genome sequencing data were obtained as previously described (Ion Torrent sequencing and assembly with CLC Genomics Workbench for Genbank accession number MT863559, 2015 [70] or with bwa for Genbank accession numbers MT863560-1, 2018 [71]). Multiple alignment of the nucleotide sequences was performed using the ClustalW algorithm and phylogenetic analysis was performed using the Neighbor–Joining and Maximum Likelihood methods in MEGA7 [36].

### 4.7. Statistical Analysis

We used negative binomial generalized linear models to analyze the mosquito trapping data. The dependent variable was the number of trapped *Cx. pipiens*, and the independent variables were the department (Bouches-du-Rhône–reference class, Gard, or Hérault) and the year (2015–2019, the reference class being 2018). Two models were separately fitted: One for the yearly total number of trapped *Cx. pipiens*, and the other for the number of *Cx. pipiens* trapped in June. Statistical analyses were performed using R 3.6.1 [72].

## Figures and Tables

**Figure 1 pathogens-09-00908-f001:**
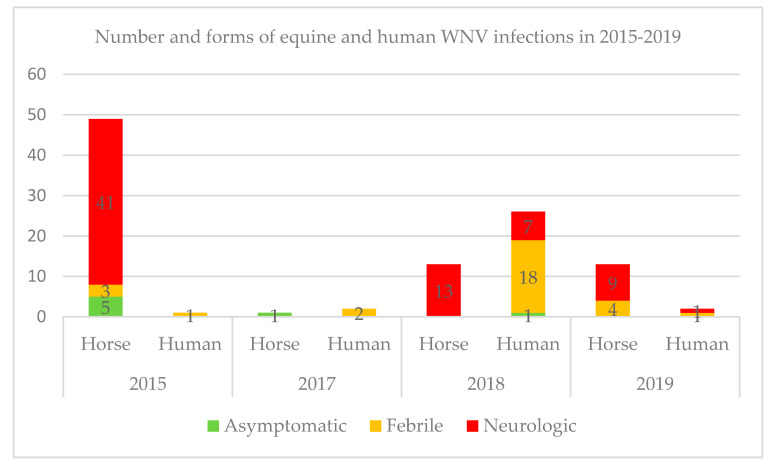
A graph that shows the total number and clinical forms of human and equine laboratory-confirmed cases per year in France from 2015 to 2019. The highest number of human and equine cases for the 2015–2019 period was reported in 2018 and 2015 respectively.

**Figure 2 pathogens-09-00908-f002:**
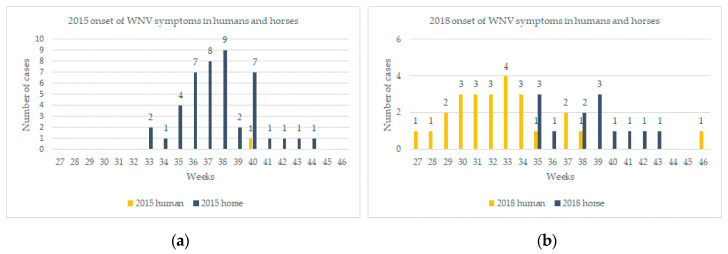
(**a**) Weekly comparison of human and equine WNV cases notifications in 2015 in France and (**b**) weekly comparison of human and equine WNV cases notifications in 2018 in France. Diagrams depict dates of onset of symptoms whenever available, and in the absence of data on symptoms onset, date of veterinary samples (five cases in 2018) or sample reporting (one case in 2018) are shown.

**Figure 3 pathogens-09-00908-f003:**
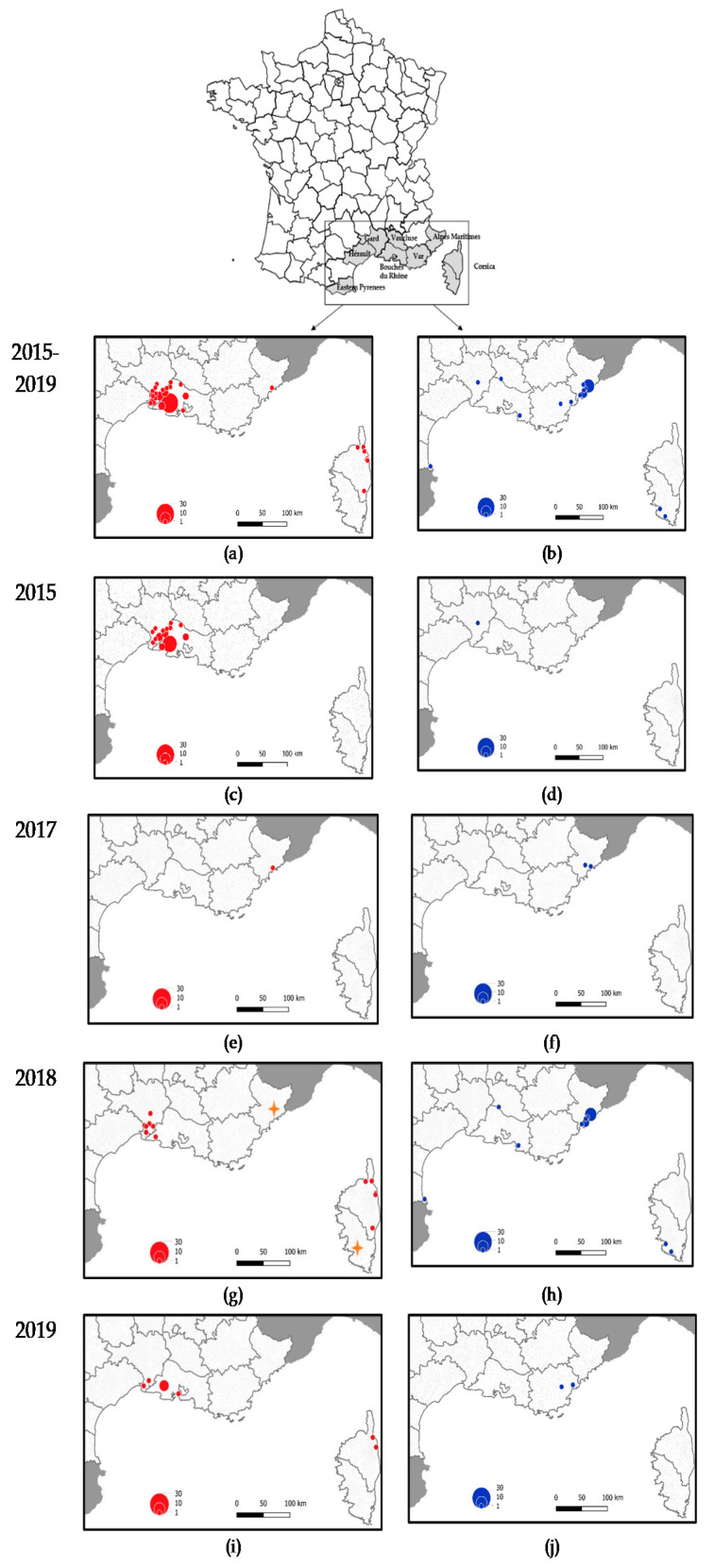
(**a**,**b**) Comparison of WNV case distribution in humans (blue dots) and horses (red dots) during 2015–2019; (**c**,**d**) Period 2015; (**e**,**f**) Period 2017; (**g**,**h**) Period 2018; and (**i**,**j**) Period 2019. In 2018, distribution of bird cases is represented by an orange star.

**Figure 4 pathogens-09-00908-f004:**
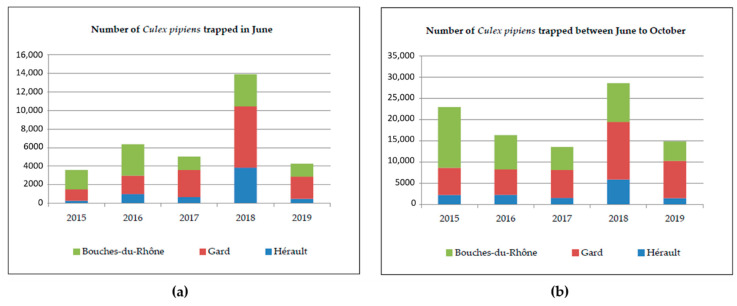
(**a**) Number of *Culex pipiens* mosquitoes trapped in three departments of the Camargue area (Bouches-Du-Rhône, Gard, and Hérault) in June during the 2015–2019 seasons and (**b**) number of *Culex pipiens* mosquitoes trapped in three departments of the Camargue area (Bouches-Du-Rhône, Gard and Hérault) between June to October during the 2015–2019 seasons.

**Figure 5 pathogens-09-00908-f005:**
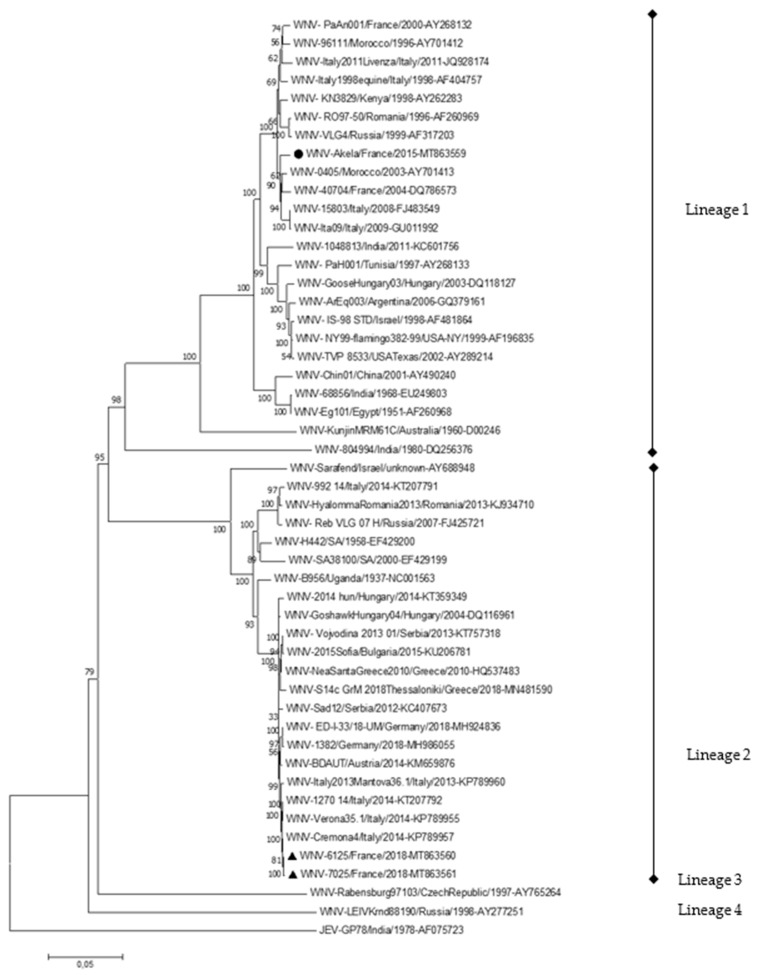
Molecular phylogenetic tree of WNV complete genome sequences detected in one horse (2015, black circle) and two birds (2018, black triangles) in France. The evolutionary history was inferred using the Neighbor–Joining and Maximum Likelihood methods in MEGA7 [36]. The optimal tree generated using the Neighbor–Joining method with the sum of branch length = 1,28228091 is shown. The percentage of replicate trees in which the associated taxa clustered together in the bootstrap test (1000 replicates) are shown next to the branches. The tree is drawn to scale, with branch lengths expressed in the same units as for the evolutionary distances used to infer the phylogenetic tree. The evolutionary distances were computed using the Jukes–Cantor method and are in the units of the number of base substitutions per site. The analysis involved 50 nucleotide sequences, including Japanese Encephalitis Virus as an outgroup. All positions containing gaps and missing data were eliminated. There were a total of 10,475 positions in the final dataset.

**Table 1 pathogens-09-00908-t001:** Negative binomial generalized linear models of the number of trapped *Culex pipiens* mosquitoes in three departments of Southern France, in June (a) or during the year (b), from 2015 to 2019.

Variable	Value	*(a) Cx. pipiens* Trapped in June		(b) Total *Cx. pipiens* Trapped	
		Abundance Ratio	*p*-Value	Coefficient	*p*-Value
Department	Bouches du Rhône	Reference		Reference	
	Gard	1.14	0.53	1.02	0.92
	Hérault	0.36	<0.0001	0.30	<0.0001
Year	2018	Reference		Reference	
	2015	0.20	<0.0001	0.66	0.054
	2016	0.40	0.0007	0.51	0.002
	2017	0.30	<0.0001	0.40	<0.0001
	2019	0.24	<0.0001	0.43	<0.0001

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
