# Peer review of "Contrasted Epidemiological Patterns of West Nile Virus Lineages 1 and 2 Infections in France from 2015 to 2019"

_pathogens, 2020, doi:10.3390/pathogens9110908_

Round 1

Reviewer 1 Report

This is an interesting publication reporting on the WNV epidemiology in France. I only have a couple of minor remarks.

Introduction:

  • Lines 49-58. Please indicate from which hosts the cases are reported from (and not only in the last line of this paragraph). To understand the WNV epidemiology it is important to know whether cases are reported from animals or humans.
  • Lines 62-67. I would keep the chronology and mention Greece 2010 before Italy 2011.

Results:

  • Line 143. Buteo buteo is the common buzzard and is not the long-eared owl. Please correct this.
  • Lines 186-196. It would be interesting to include samples from the different years (if available) to see when Lineage 2 actually emerged in France.
  • Line 208-211. This is a bit a strange statement. WNV-Akela/France/2015 and WNV-XXXX/France/2018 samples belong to different lineages, so they are different. The difference within the lineages of samples collected at different time points is important however.
  • Line 249. The abbreviation of the European Centres for Diseases Prevention and Control is ECDC (with capital E).

Discussion:

  • Line 218. Would be good to reiterate that WNV is not new for France and that it has been detected in end 1950’s and in the 1960’s.

Methods:

  • Lines 382. Why did you opt for a NJT, I would suggest to perform this analysis using a ML method which will provide a more robust estimate/tree.
  • Lines 385-390. Species names should be in italics. Please check the entire publication
  • Figure 3. The figure is not of very good quality. Please increase the resolution. It would be good to indicate which part of France is shown. Not everybody is familiar with the geography of France. And indicate the place names you mention in the text.

Author Response

Author's Response: Please see the attachment.

Reviewer 2 Report

The authors report on recent French WNV outbreaks (2015-2019) in humans, equids and the wild avifauna in the Mediterranean area, providing data of interest especially for epidemiologists. I found the manuscript quite clear and well written. I have some comments/suggestions which I list below.

Major Points

  • Statistical analysis. Why NB and not Poisson? You might consider to add some histograms showing the captures distribution (as supplementary would be fine).
  • Why the focus on June and not for instance whole summer?

Minor Points

  • Line 47: either strains or lineages.
  • Line 48: in the 1960s.
  • Figure 2. There seems to be a shift in infection patterns, from humans to horses. Any explanation? Possibly a shift in the vector feeding preference? Please discuss.
  • Figure 3. Corsica is quite interesting: horses are infected in the North but human infections occur in the South, where positive birds were found. How would you explain the lack of human infections in the Northern part?
  • Lines 285-286: Indeed I agree. Some remarks:
    - I wouldn't say "even if" because actually a lower circulation in 2019 might be explained but the high circulation of the previous year, which could have led to a larger immunity in the reservoir population (birds).
    - According to ECDC surveillance atlas some WNV autochthonous cases have been recorded in newly affected areas in Germany this year (2020). So indeed, WNV epidemiology is expected to change more generally over Europe.
  • Lines 306-208. Can you add a reference for the Usutu outbreaks?

Author Response

(The authors gave the same response as above.)
